# Saliva Diagnostics in Spaceflight Virology Studies—A Review

**DOI:** 10.3390/v16121909

**Published:** 2024-12-12

**Authors:** Douglass M. Diak, Brian E. Crucian, Mayra Nelman-Gonzalez, Satish K. Mehta

**Affiliations:** 1Aegis Aerospace, Human Health and Performance Directorate, Houston, TX 77058, USA; douglass.m.diak@nasa.gov; 2National Aeronautics and Space Administration (NASA) Johnson Space Center, Human Health and Performance Directorate, Houston, TX 77058, USA; 3KBR, Human Health and Performance Directorate, Houston, TX 77058, USA; mayra.a.nelman@nasa.gov; 4JES Tech, Human Health and Performance Directorate, Houston, TX 77058, USA

**Keywords:** saliva, spaceflight, herpesviruses, International Space Station, biomarkers, immune

## Abstract

Many biological markers of normal and disease states can be detected in saliva. The benefits of saliva collection for research include being non-invasive, ease of frequent sample collection, saving time, and being cost-effective. A small volume (≈1 mL) of saliva is enough for these analyses that can be collected in just a few minutes. For “dry” saliva paper matrices, additional drying times (about 30 min) may be needed, but this can be performed at room temperature without the need for freezers and specialized equipment. Together, these make saliva an ideal choice of body fluid for many clinical studies from diagnosis to monitoring measurable biological substances in hospital settings, remote, and other general locations including disaster areas. For these reasons, we have been using saliva (dry as well as wet) from astronauts participating in short- and long-duration space missions for over two decades to conduct viral, stress, and immunological studies. We have also extended the use of saliva to space analogs including bed rest, Antarctica, and closed-chamber studies. Saliva is a biomarker-rich and easily accessible body fluid that could enable larger and faster public health screenings, earlier disease detection, and improved patient outcomes. This review summarizes our lessons learned from utilizing saliva in spaceflight research and highlights the advantages and disadvantages of saliva in clinical diagnostics.

## 1. Introduction

Saliva has been used for the detection of biological markers for a wide spectrum of normal and disease states in humans for a long time. It is a non-invasive, easily accessible, and self-collected body fluid that contains a variety of measurable biological substances. While mostly water, saliva also contains ions, carbohydrates, proteins and peptides, exfoliated cells, nucleic acids, and microorganisms [1]. This complex set of ingredients plays a critical role in maintaining oral health. However, it can also serve as a window into systemic health when circulating blood analytes transport across the blood–saliva barrier (BSB) and mix with normal excretions of saliva produced by the saliva glands. Indeed, the concentration of many biological markers may be low or variable because permeation of the BSB occurs largely by passive, paracellular diffusion [2], although active transport across epithelial membranes has also been described [3]. Additional variations in biomarker analysis may arise if the salivary flow rate or diurnal variations are not accounted for. In any case, with proper sampling controls, these salivary analytes can provide insights into systemic emotional, hormonal, immunological, neurological, microbiological, nutritional, and metabolic states [4]. Even therapeutic and non-therapeutic drugs including psychotropic drugs, chemotherapeutic agents [5], and illicit substances like marijuana, cocaine, and alcohol [4,6] may be monitored in saliva. Furthermore, a variety of candidate salivary biomarkers are being discovered to assist in screening and diagnosing oral diseases like gingivitis [7] or oral squamous cell carcinoma [8]. As such, saliva is a multifaceted body fluid that not only provides information about local health but may also mirror that of systemic profiles.

Even with all these significant advantages, saliva has been generally underutilized in clinical research and diagnostics. However, due to the COVID-19 pandemic and large-scale viral testing, saliva collections have become common practice globally. Initially, population screenings were largely gathered by nasopharyngeal swabs, but these were uncomfortable and difficult to perform, especially for less than cooperative subjects, like children. Not surprisingly, suboptimal swab collections were a major contributor to false-negative COVID-19 test results [9]. Fortunately, research has shown that saliva-based diagnostics, which are easier to collect, have comparable sensitivity and specificity to nasopharyngeal swabs for viral testing [10]. While nasopharyngeal swabs may still be the gold standard, saliva testing has been widely adopted for COVID-19 testing [11]. As such, the medical community only recently realized the significance of using saliva as an easier means of screening for viral positivity (i.e., SARS-CoV-2) in the population. However, saliva has been used for decades supporting virology investigations at NASA to screen for viruses and, particularly, the reactivation of latent herpesviruses [12,13,14,15,16,17,18,19,20,21]. Herein, this review will highlight the advantages and disadvantages of saliva in clinical diagnostics and summarize our lessons learned from utilizing saliva in spaceflight viral research.

## 2. Advantage/Disadvantage of Saliva over Plasma or Serum

Saliva has many advantages over blood as a research and clinical diagnostic tool. Its collection, storage, and shipment (if necessary) are straightforward and economical. For patients, particularly those with trypanophobia, the non-invasive collecting technique significantly alleviates anxiety and discomfort [22], likely providing a truer analytical value than one riddled with additional biological stress responses. Unlike phlebotomy, which requires specialized equipment, storage reagents, and trained personnel, saliva collection can be performed by the patient via a simple passive-drooling procedure (unstimulated), chewing a paraffin pellet (stimulated), or saturation of a synthetic swab held in the cheek area for a few minutes. Although unique passive-drool and saliva swab storage containers have been developed to simplify downstream processes, no specific equipment is essential, and saliva can be collected and frozen in a simple tube (i.e., 1–50 mL conical tube). In contrast to blood collections, when repeated sampling is required, an indwelling catheter may be placed or repeated venipunctures occur with a needle. Both of these procedures increase the risk of secondary infections, which is not only harmful for the patient but also to the results for the analyte of interest. Saliva collections simplify this process, allowing for the gathering of repeated samples for longitudinal monitoring over time.

While usually not a perfect match, numerous studies have shown correlations between serum/plasma and saliva levels. In addition to cross-sectional sampling, both diurnal and monthly profiles of salivary hormone levels are comparable to systemic serum patterns [23] and can be used to monitor fertility cycles, menopausal fluctuations, and stress [24]. Recent research has revealed that immunoglobulins, particularly IgG but also IgA and IgM, are significantly correlated between saliva and blood [25,26]. With recent technological advances in proteomics, over 2000–3000 salivary proteins and peptides [27,28] have been discovered to mirror that of their circulating counterparts. However, while this roughly 30% overlap is respectable, analytes in saliva are generally present in lower amounts or absent completely compared to blood [4]. Thus, it is important for the researcher or clinician to know the limit of detection in saliva for the analyte of interest. Conversely, having too much of an analyte via blood contamination or local oral inflammation can produce inaccurate results with false positives or extremely high levels. Patients with oral pathologies, like gingivitis or periodontitis, are generally excluded from saliva research, with the exception of diagnosing that specific encumbrance. Furthermore, while saliva does not clot like blood samples, inadequate hydration, smoking, or excess mucus from allergies can make it quite thick and viscous, proving technically difficult to manipulate in the lab.

Blood collection tubes (BCTs) also already contain a preservative to maintain sample integrity for a period of time. Depending upon the analyte of interest, BCTs can also be refrigerated or frozen to help reduce the breakdown of the sample. For raw saliva, it is essential to freeze the sample as quickly as possible to maintain sample integrity (usually protein analysis). Refrigeration for up to 2 h is also nominal, when immediate freezing is not an option. This helps minimize degradation of the sample by enzymatic activity via amylases, proteases, and lysozymes while also reducing microbial growth and contamination [29]. Thus, it is important to mitigate the effects of these enzymes and preserve the integrity of the samples for scientific analysis. Some have tested protocols utilizing protease inhibitor cocktails to protect from saliva protein degradation, but the results from these tests are mixed [30,31]. Others have developed DNA preservation buffers that when added to the saliva sample may preserve DNA integrity for up to 24 months when stored at room temperature [32]. In our studies, we found that simply freezing the sample immediately (−20 °C or lower) and then limiting downstream freeze/thaw cycles by initial sample aliquoting produces the best results.

## 3. Saliva for Viral Studies

Saliva has been reliably used to detect HIV 1 and 2, viral hepatitis A, B, and C [4], and many human herpes viruses including EBV, CMV, HSV 1 and 2, VZV, and HHV6. Some of these herpesviruses (alpha herpes viruses, like VZV and HSV1) are neurotropic (reside in the nervous system), but recent studies have shown that they can be found in saliva as well [33]. During our spaceflight studies at Johnson Space Center, NASA, we used saliva to detect herpes virus DNA by real-time quantitative PCR. These studies were carried out in ground-based space analogs including Antarctica [34], bed rest studies, NASA’s underwater studies (NEEMO), and artificial gravity models, as well as in short- and long-duration space missions [21]. Epstein–Barr virus (EBV), a gamma herpes virus, one of the most common latent herpes viruses, reactivated and shed in larger quantities coincidentally with decreasing immunity [35,36] in the saliva of the subjects in these studies. These studies were also expanded to include other herpes viruses, like cytomegalovirus (CMV) [37] and HSV1 [38]. Unsurprisingly, these viruses also showed similar trends, in that spaceflight stress and immune dysregulation induced a reactivation of latent viral replication, including some cases of adverse clinical events where medical intervention was needed. In addition, we also studied varicella zoster virus (VZV), a neurotropic alpha herpesvirus. Its primary infection usually causes varicella (chicken pox), after which the virus becomes latent in ganglionic neurons along the entire neuroaxis [39,40]. Later in life, it can reactivate to produce zoster (shingles). In some cases, it can be followed by chronic pain (postherpetic neuralgia [PHN]), mostly in individuals 60 years of age or older. We reported that spaceflight-associated stress resulted in the reactivation and shedding of VZV in the saliva of otherwise healthy astronauts asymptomatically during and after the flight that later on was also shown to be a live and infectious virus [14] which would have a potential risk of infecting seronegative individuals that were on board. These findings highlight the significant risk of latent herpes viral reactivations and shedding in astronauts during and after spaceflight and the need for directed immune countermeasures (see review for more information [41]).

We are also utilizing and innovating with “dry saliva” technologies for herpes viral assessment. This technology is similar to what we developed for stress hormones (see the Saliva for Stress Hormones Section for more details). Dry saliva is uniquely suitable for deep space mission constraints (i.e., to the Moon and Mars) due to low volume, no processing requirement, and no conditioned storage (i.e., freezer). As opposed to raw saliva collections which entail a passive-drool of saliva in a volumetric container or saturating an oral swab and then freezing the sample, dry saliva collections utilize specialized filter paper materials that maintain sample integrity upon drying at room temperature. The storage of dried saliva matrices, which requires unconditioned stowage, is energy-, space-, and cost-effective, making this an excellent tool for collection when extreme sample-to-lab issues arise. This collection process is similar to that of “dry blood spots” which have been standardized and used extensively in many diagnostic applications, especially for newborns [42]. The paper matrix is saturated either directly by mouth or, if the paper has a chemical preservative applied to it to enhance sample integration, by separate application by pipette from a pooled raw saliva sample. The paper is then dried at room temperature and stored in a unique stabilization bag with a desiccant until processing in the lab. Potential limitations to these methods arise from the need for specialized pretreatment or rehydration protocols of the dry paper matrices for less stable analytes. Current research in our lab is optimizing protocol requirements for about 80 different analytes for dry saliva diagnostics (unpublished to date). Filter paper type (i.e., with or without chemical additives) and quantities depend on downstream applications, but a variety of matrices have been created to ensure diagnostic analyte integrity, including those for viral DNA/RNA and protein analysis as well as chemistry biomarkers. The ultimate aim of this technology is to achieve results in our target analytes which are comparable to frozen samples.

Our spaceflight-developed technologies of saliva collection in remote locations, even without gravity, storing at room temperature, and processing for rapid viral detection, have been applied to diagnose various clinical conditions including acute zoster [43], zoster sine herpete [44], chickenpox [45], and postherpetic neuralgia (PHN) [46]. Viral detection has been demonstrated even before the rash appears, which now makes diagnosis less invasive and less time-consuming. In fact, a rapid and sensitive virus detection method has been developed and used to detect virus in saliva samples taken from asymptomatic patients with neurologic and other VZV-related disease [45], multiple sclerosis [47], and various other neurological disorders [48,49]. These protocols are being employed in various clinics and hospitals, including the CDC and Columbia University in New York, as well as in Switzerland and Israel.

## 4. Saliva for Stress Hormones

Because of our unique research environment and the link between stress, immunity, and viral reactivation, we devised a collection tool for the collection of human saliva samples for stress hormone assessments. This tool can be used at any remote location, including space, and does not require any special conditions for storage. Saliva samples are collected on filter paper and dried at room temperature. We developed a packaging appliance for these filters that was compact and easy to use, specifically for NASA, called the Saliva Procurement and Integrated Testing (SPIT) booklets (Figure 1). These booklets are being used to collect and store saliva samples from International Space Station crewmembers before, during, and after the spaceflight.

Using this specialized SPIT booklet, astronauts could take multiple samples throughout the day (from waking to right before sleeping), dry the entire booklet at ambient temperature, store it in specialized bags, and return them to Earth for analysis. This is a novel method of the collection of saliva samples that is non-invasive, inexpensive, does not require trained medical personnel, and can be stored at room temperature for at least 6 months without compromising the quality of the sample’s stress hormone assessment. Following sample extraction methods from these filters (published methods here [23]), steroid hormones can then be measured using any commercially available, high-sensitivity kit. The results are reliable, consistent, and correlate very well with the blood measurements.

This innovation has allowed for studies to be performed on the effects of spaceflight (short: 10–16 days and long: up to 1 year) on cortisol and DHEA levels, two important stress and immune regulatory hormones, on multiple spaceflight timepoints, as well as the study of the effects of the circadian rhythm on these hormones in this unique environment [19,20]. Cortisol regulates the body’s stress response and helps suppress inflammation, while DHEA counterbalances cortisol by enhancing immunity and promoting anabolic effects. The molar ratio of cortisol to DHEA [C]/[D] is thus often used as a vital marker of immune regulation and stress response homeostasis. Using the SPIT booklets in these flight studies, we analyzed the regular diurnal release of these hormones through the various phases of flight: pre-, in-, and post-flight [20]. The diurnal patterns of salivary cortisol were elevated during flight, while DHEA was reduced. Thus, we have observed through this saliva collection technology an increased [C]/[D] ratio during spaceflight (Figure 2), suggesting potential challenges to the immune system while astronauts are in space. The salivary hormone data from these studies have been linked to a broader immune modulation [50], including increased inflammatory cytokine response and TH2 shift observed in plasma samples from earlier spaceflight studies [18,51].

## 5. Saliva for Immunology Biomarkers

Saliva samples can be very informative, for research or clinical purposes, for a variety of immune system biomarkers. Cytokine concentration is an excellent indicator of the hormonal regulation of immunity, inflammation, Th1/2 shifts, and other processes. We have previously published saliva cytokine alterations in ISS crewmembers [17]. In congruence with a study performed on healthy subjects [52], we observed correlations between salivary and plasma IL-6 and IFN-γ in astronaut samples before, during, and after spaceflight. Williamson et al. further reported correlations with MIP-1β, while our study in astronauts included correlations in IL-8 and IL-12p70. The relatively low number of correlations (5/27) between saliva and plasma is not unexpected, however, considering the study populations in these studies: healthy adults and highly selected, physically fit astronauts. Given the abundance of IL-6, IFN-γ, MIP-1β, and IL-8 in normal physiological states, the correlations between the saliva and plasma of these cytokines are logical. Conversely, low to normal levels of other remote cytokines in plasma may not consistently and passively diffuse across the BSB and show up regularly in saliva. The successful measurement of at least 27 cytokines in these studies provides validation that saliva could be used as a platform for both the assessment and longitudinal tracking of cytokine levels in both research and clinical disease studies. Furthermore, in another study on ISS crewmembers and ground controls, saliva was used to assess a variety of first-line innate immune antimicrobial defenses [12]. These findings from saliva analysis furthered the evidence that spaceflight can impact the immune system of astronauts. In addition, we have established protocols that enable the collection, storage, and recovery of saliva cytokines for up to 1+ years (unpublished data). Other potential targets for saliva monitoring include antibody concentrations (i.e., SARS-COV-2 anti-spike Igs [53]), both generic and pathogen-specific, antimicrobial proteins, and general markers of inflammation (CRP, etc.).

## 6. Saliva for Other Biomarkers

Saliva has the potential to be valuable in monitoring other types of biomarkers. These include markers of bone function and markers of nutritional status. Cardiovascular disease may be informed by measuring CKMB or myoglobin. Measurements of HbA1C can give information about diabetes risk. Leptin and ghrelin are hormones produced by the salivary glands that are involved with the regulation of body weight, energy balance, and glucose metabolism [54,55]. A variety of growth factors like insulin-like growth factor 1 (IGF-1) and epidermal growth factor (EGF) have also been measured in saliva, indicating their use as biomarkers of bone, tissue, and cellular synthesis [56]. Heat shock protein 70 (HSP70) is a protein that is also detected in saliva and involved in the signaling networks for immune cells to release pro-inflammatory cytokines. This protein is a stress-induced protein involved in several pathological conditions, including autoimmune diseases such as arthritis, dermatitis, Guillan–Barre syndrome, and systemic lupus; the use of saliva samples would serve as a non-invasive, cost-effective method for the detection of some of these conditions [57]. Heat shock protein 47 (HSP47), which is involved in the regulation of collagen maturation, is also detected in the saliva [58]. Infectious diseases may be informed not just by the measurement of virus DNA but by the detection of a wide variety of other pathogens, including bacteria or fungi. Further, it may be possible to develop the isolation and characterization of intact cells, either endothelial (latent virus DNA, as we have previously published) or immune cells (neutrophils, lymphocytes, etc.), to inform other types of physiological dysregulation or disease processes.

## 7. Conclusions

To illustrate the advantage of using saliva in clinical research, in one recent study we conducted [59], saliva was used to determine the effectiveness of the prophylactic administration of an antiviral drug (valacyclovir) as a countermeasure to herpesvirus shedding and reactivation in Antarctic expeditioners. Briefly, the subjects were either given valacyclovir or a placebo every day for the duration of their mission (6–8 months). The viral shedding, in terms of both viral load and frequency, was remarkably reduced for both EBV and HSV-1 in the treatment group, while the reactivation of VZV was not detected in any sample. Additionally, saliva supernatants were used to assess traditional stress markers (DHEA, cortisol, and alpha-amylase) and the 13-plex cytokine immune profile of the expeditioners. Essentially, no difference was seen in the stress hormones, whereas many cytokines were suppressed. Further, additional immunological markers are currently being assessed with the remaining frozen volumes (unpublished to date). This experiment truly encompasses the unique benefit that analyzing saliva can have for diagnostic purposes. For instance, for 6–8 months at stations in Antarctica (McMurdo or South Pole), a saliva sample was donated for 5 straight days every month by over 30 individual subjects, highlighting its ease of use in a longitudinal study. The efficiency of the sampling process maintained subject adherence to the protocols and limited study dropouts. Over a thousand samples were easily collected, stored, and frozen by the subjects until shipment by the study coordinators to Johnson Space Center, limiting the need for specialized equipment and personnel. A wide variety of analytes were assessed from each sample, covering three different scientific disciplines of virology, stress, and immunology. While blood sampling may have been feasible insofar as budget, time, expertise, and resources allow, it would not have been as easy and inexpensive as saliva while covering fundamentally the same analyses.

There is clearly more to saliva than its physical properties of moistening and protecting the oral cavity. Overall, saliva represents an important physiological fluid that comprises a complex mixture of analytes, including viruses, enzymes, hormones, nucleic acids, proteins, immune markers, minerals, and cells. Beneficially, these secretions into saliva can also be used for biomarker detection in clinical diagnostics for not only virology but a wealth of other scientific disciplines. Its non-invasive and simplistic self-collection procedures make it an attractive stress-free diagnostic fluid with minimal processing requirements (both in liquid and dry storage form) to study a variety of normal and disease states in spaceflight research and beyond.

## Figures and Tables

**Figure 1 viruses-16-01909-f001:**
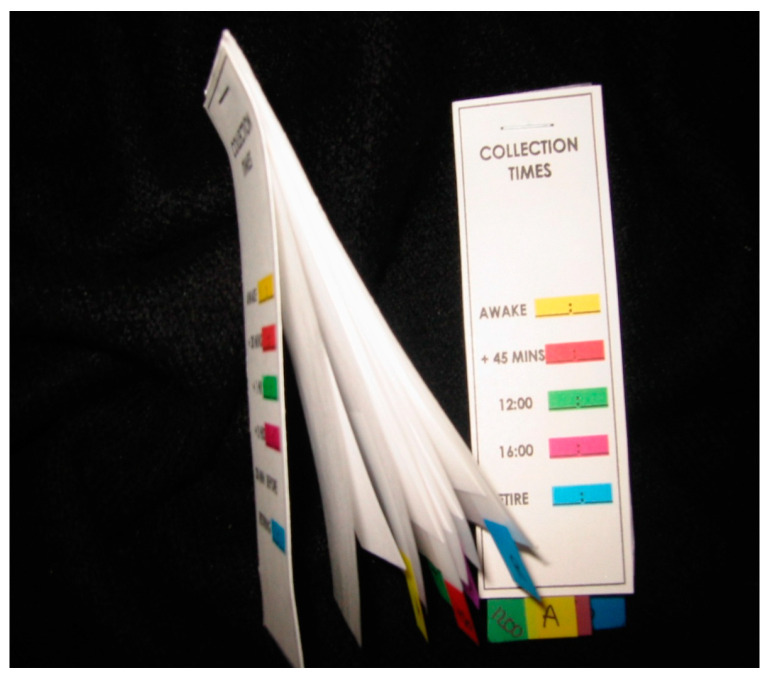
Saliva Procurement and Integrated Testing (SPIT) booklet.

**Figure 2 viruses-16-01909-f002:**
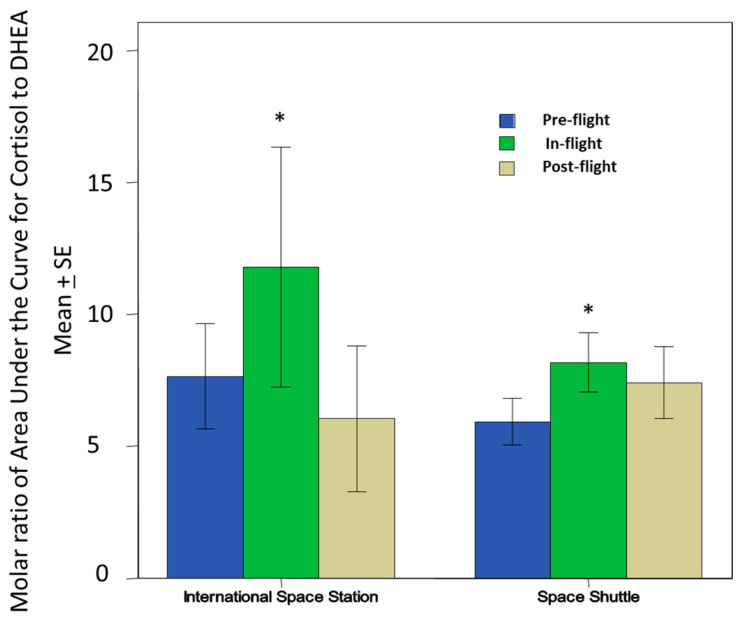
Salivary cortisol to DHEA [C]/[D] ratio comparison between pre-, in-, and post-flight timepoints within the International Space Station and the Space Shuttle eras. There is a significant increase in that ratio during flight for both Space Shuttle (N = 17) or ISS (N = 10). This increase may be associated with lower cellular immunity and innate immunity. This could also contribute to potentially greater inflammatory cytokines that would affect bone remodeling and bone growth. (Image taken from [21] Copyright © 2019 Rooney, Crucian, Pierson, Laudenslager and Mehta.), * *p* < 0.01.

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
