# Peer review of "Saliva Diagnostics in Spaceflight Virology Studies—A Review"

_viruses, 2024, doi:10.3390/v16121909_

Round 1
Reviewer 1 Report
Comments and Suggestions for Authors
Manuscript ID: viruses-3318460
general comment:
the manuscript is a well detailed review description of dry saliva uses for diagnostic.
Moreover, in my opinion, it appropriately covers the advantages of using saliva, mainly dry specimens for spaceflight diagnostics.
I would suggest a very limited changes/upgrading.
Abstract
Just add a very short sentence regarding saliva volumes and drying processes. Its importance considering the spaceflight applications.
Introduction
I suggest including a comment regarding blood/circulation/saliva district relationships/exchanges. Moreover, the importance of mouth salivary local compound synthesis.
Maybe, I suggest the inclusion of a reference regarding virology investigation at NASA.
Advantages/Disadvantages
This paragraph is very well described. Possibly include some disadvantages .. i.e. enzyme action (amylases, proteases, ..) in the storage of salivary fluids.
Salivary for viral studies; Salivary for stress hormones
These paragraphers are well appropriate. Very good.
Saliva for immunology biomarkers
Please include a literature reference regarding the last sentence regarding antibody concentrations.
Saliva for other biomarkers
Regarding hormone production by salivary cells, please include growth factors (i.e. IGF). Furthermore, consider the biomarkers regarding the wound repair and salivary restoration effects.
Conclusions
In the last sentence …
“Overall, … including viruses, enzymes, hormones, nucleic acids, …”
Author Response
Reviewer #1 Comments and Response
general comment:
the manuscript is a well detailed review description of dry saliva uses for diagnostic.
Moreover, in my opinion, it appropriately covers the advantages of using saliva, mainly dry specimens for spaceflight diagnostics.
We thank the reviewer for kind words.
I would suggest a very limited changes/upgrading.
Abstract
Just add a very short sentence regarding saliva volumes and drying processes. Its importance considering the spaceflight applications.
We thank the reviewer for the suggestion and agree the minimal volume and drying process is important for spaceflight applications. We have added 2 sentences to the abstract to briefly highlight these assets of saliva.
Introduction
I suggest including a comment regarding blood/circulation/saliva district relationships/exchanges. Moreover, the importance of mouth salivary local compound synthesis.
Maybe, I suggest the inclusion of a reference regarding virology investigation at NASA.
We thank the reviewer for the suggestion and agree that information about the blood-saliva barrier would inform the reader of the mechanism of action in how systemic analytes can be assessed in saliva. We’ve also included a statement about local saliva analysis and references to that information. As suggested, we have also included a host of NASA virology investigations that utilize saliva for viral research throughout the years.
Advantages/Disadvantages
This paragraph is very well described. Possibly include some disadvantages .. i.e. enzyme action (amylases, proteases, ..) in the storage of salivary fluids.
We thank the reviewer for those kind words and have expanded on our paragraph about sample preservation to include some specific references to amylases, proteases, and lysozymes.
Salivary for viral studies; Salivary for stress hormones
These paragraphers are well appropriate. Very good.
We thank the reviewer for those kind words.
Saliva for immunology biomarkers
Please include a literature reference regarding the last sentence regarding antibody concentrations.
We have added a context reference to the use of saliva to measure antibody concentrations as suggested.
Saliva for other biomarkers
Regarding hormone production by salivary cells, please include growth factors (i.e. IGF). Furthermore, consider the biomarkers regarding the wound repair and salivary restoration effects.
We have added a sentence highlighting growth factors like IGF as well, as we agree, these are also major hormones measured in saliva.
Conclusions
In the last sentence …
“Overall, … including viruses, enzymes, hormones, nucleic acids, …”
We have added “hormones” to the sentence as suggested.
Reviewer 2 Report
Comments and Suggestions for Authors
The manuscript titled "Saliva in Spaceflight Virology Studies - A Review" presents a comprehensive and informative review of the use of saliva as a diagnostic tool, particularly in the unique environment of spaceflight. The review effectively highlights the advantages, limitations, and applications of saliva-based diagnostics, with a focus on viral and stress-related biomarkers. It is well-structured and covers relevant literature, providing a balanced discussion. Overall, the paper is scientifically sound and well-written. However, minor revisions are recommended to improve clarity, consistency, and flow.
Major COMMENTS:
1. The title is appropriate and engaging, but consider adding “diagnostics” to make it more precise.
2. The abstract is clear and concise but could benefit from a concluding sentence summarizing the potential implications for future research or clinical diagnostics.
3. The introduction is comprehensive, but it would be helpful to include a brief mention of the limitations of using saliva (e.g., lower analyte concentrations) earlier in the section for balance.
4. The transition to the discussion of spaceflight virology research could be more explicit to better guide the reader.
5. In the Section "Saliva for Viral Studies", the authors provide a rich discussion of herpesviruses, but the focus on EBV and VZV could be expanded to briefly compare with findings for other latent viruses such as CMV or HSV-2.
AND, the mention of “dry saliva” technology is intriguing but could use additional context regarding the potential limitations or technical challenges in standardizing such methodologies for broader use.
6. The use of the SPIT booklet is well-described and novel. However, the paper would benefit from including a figure or diagram of this device to aid visualization.
7. For cytokines, a comparison with blood cytokine profiles (e.g., whether saliva offers equivalent or complementary insights) would strengthen the discussion.
8. Providing a summary table listing biomarkers, their analyte type, and corresponding applications in spaceflight or terrestrial diagnostics could enhance readability.
Author Response
Reviewer #2. Comments and Response
The manuscript titled "Saliva in Spaceflight Virology Studies - A Review" presents a comprehensive and informative review of the use of saliva as a diagnostic tool, particularly in the unique environment of spaceflight. The review effectively highlights the advantages, limitations, and applications of saliva-based diagnostics, with a focus on viral and stress-related biomarkers. It is well-structured and covers relevant literature, providing a balanced discussion. Overall, the paper is scientifically sound and well-written. However, minor revisions are recommended to improve clarity, consistency, and flow.
We thank the reviewer for those kind words.
Major COMMENTS:
- The title is appropriate and engaging, but consider adding “diagnostics” to make it more precise.
We have adjusted the title to include “diagnostics”. We agree with the reviewer its addition makes it more precise.
- The abstract is clear and concise but could benefit from a concluding sentence summarizing the potential implications for future research or clinical diagnostics.
We thank the reviewer for the comment on the abstract. We have included a summary sentence to prior to the description of the review to help finalize the point.
- The introduction is comprehensive, but it would be helpful to include a brief mention of the limitations of using saliva (e.g., lower analyte concentrations) earlier in the section for balance.
We thank the reviewer for the comment on the introduction. We have added a few limitations of using saliva to the introductory paragraph to improve the balance of the writing.
- The transition to the discussion of spaceflight virology research could be more explicit to better guide the reader.
We agree with the reviewer. A proper transition of saliva in virology testing was lacking. We have adjusted the second paragraph in the introduction to better synchronize the two ideas and guide the reader.
- In the Section "Saliva for Viral Studies", the authors provide a rich discussion of herpesviruses, but the focus on EBV and VZV could be expanded to briefly compare with findings for other latent viruses such as CMV or HSV-2.
We have added a brief summary sentence to add information about CMV and HSV-1 in spaceflight. We also have added a conclusion sentence to the end of this paragraph that summarizes the effect of spaceflight on all these herpesviruses because, in general, they act the same in spaceflight (i.e. latent viral reactivation due to stress and immune dysregulation).
AND, the mention of “dry saliva” technology is intriguing but could use additional context regarding the potential limitations or technical challenges in standardizing such methodologies for broader use.
We thank the reviewer for the suggestion and agree that the dry saliva technology does have potential limitations. We have added our thoughts to the paragraph on dry saliva to discuss the limitations of this technology and its standardization for broader use.
- The use of the SPIT booklet is well-described and novel. However, the paper would benefit from including a figure or diagram of this device to aid visualization.
The SPIT booklet is presented in Figure 1.
- For cytokines, a comparison with blood cytokine profiles (e.g., whether saliva offers equivalent or complementary insights) would strengthen the discussion.
We thank the reviewer for the suggestion and agree additional comparisons between plasma and saliva cytokines was needed to strengthen the argument. We have added multiple sentences to this section to give the reader more insight into our thinking.
- Providing a summary table listing biomarkers, their analyte type, and corresponding applications in spaceflight or terrestrial diagnostics could enhance readability.
While we agree with the reviewer that a summary table listing all the salivary biomarkers, their analyte type, and corresponding applications used in spaceflight and terrestrial diagnostics would simplify the written paragraphs, we believe the paragraphs are sufficient in providing the reader with a wealth of information and references.